# Genetically Engineered Bacterial Ghosts as Vaccine Candidates Against *Klebsiella pneumoniae* Infection

**DOI:** 10.3390/vaccines13010059

**Published:** 2025-01-10

**Authors:** Svetlana V. Dentovskaya, Anastasia S. Vagaiskaya, Alexandra S. Trunyakova, Alena S. Kartseva, Tatiana A. Ivashchenko, Vladimir N. Gerasimov, Mikhail E. Platonov, Victoria V. Firstova, Andrey P. Anisimov

**Affiliations:** 1Laboratory for Plague Microbiology, Especially Dangerous Infections Department, State Research Center for Applied Microbiology and Biotechnology, 142279 Obolensk, Russia; dentovskaya@obolensk.org (S.V.D.); vagaiskaya.anastasiya@gmail.com (A.S.V.); sasha_trunyakova@mail.ru (A.S.T.); platonov@obolensk.org (M.E.P.); 2Laboratory of Molecular Biology, State Research Center for Applied Microbiology and Biotechnology, 142279 Obolensk, Russia; kartseva_as@mail.ru (A.S.K.); ivaschenko_ta@mail.ru (T.A.I.); firstova@obolensk.org (V.V.F.); 3Department of Disinfectology, State Research Center for Applied Microbiology and Biotechnology, 142279 Obolensk, Russia; ilcvngerasimov@obolensk.org

**Keywords:** *Klebsiella pneumoniae*, mice, inactivated vaccine, holin–endolysin system, bacterial ghost, protection

## Abstract

**Background/Objectives** Bacterial ghosts (BGs), non-living empty envelopes of bacteria, are produced either through genetic engineering or chemical treatment of bacteria, retaining the shape of their parent cells. BGs are considered vaccine candidates, promising delivery systems, and vaccine adjuvants. The practical use of BGs in vaccine development for humans is limited because of concerns about the preservation of viable bacteria in BGs. **Methods:** To increase the efficiency of *Klebsiella pneumoniae* BG formation and, accordingly, to ensure maximum killing of bacteria, we exploited previously designed plasmids with the lysis gene *E* from bacteriophage φX174 or with holin–endolysin systems of λ or L-413C phages. Previously, this kit made it possible to generate bacterial cells of *Yersinia pestis* with varying degrees of hydrolysis and variable protective activity. **Results:** In the current study, we showed that co-expression of the holin and endolysin genes from the L-413C phage elicited more rapid and efficient *K. pneumoniae* lysis than lysis mediated by only single gene *E* or the low functioning holin–endolysin system of λ phage. The introduction of alternative lysing factors into *K. pneumoniae cells* instead of the E protein leads to the loss of the murein skeleton. The resulting frameless cell envelops are more reminiscent of bacterial sacs or bacterial skins than BGs. Although such structures are less naive than classical bacterial ghosts, they provide effective protection against infection by a hypervirulent strain of *K. pneumoniae* and can be recommended as candidate vaccines. For our vaccine candidate generated using the O1:K2 hypervirulent *K. pneumoniae* strain, both safety and immunogenicity aspects were evaluated. Humoral and cellular immune responses were significantly increased in mice that were intraperitoneally immunized compared with subcutaneously vaccinated animals (*p* < 0.05). **Conclusions:** Therefore, this study presents novel perspectives for future research on *K. pneumoniae* ghost vaccines.

## 1. Introduction

*Klebsiella pneumoniae* can be isolated from sewage, polluted waters, and soils, but it also behaves as a representative of the normal flora of the gastrointestinal tract of humans and other animals [1]. In other organs of the host, it becomes an opportunistic pathogen capable of causing a number of diseases, including pneumonia, bacteremia, soft tissue infections, meningitis, and urinary tract infections (sometimes leading to death) [2]. Multidrug resistance in *K. pneumoniae* renders antibiotic therapy ineffective, highlighting the need for the development of *Klebsiella* vaccines. But here, researchers face certain difficulties. The serological typing of capsular antigens (K-antigens) or lipopolysaccharide’s (LPS) O-specific polysaccharide chains (O-antigens) has established the presence of 82 capsular and 9 O-antigen types [3]. LPS serovars O1, O2, and O3 are the most common and are responsible for approximately 80% of all *Klebsiella* infections. Capsular serovars K1 and K2 contain hypervirulent strains of various O serovars. Currently, there is no licensed vaccine against *K. pneumoniae* infection. Various types of Klebsiella vaccine candidates (whole-cell vaccines, outer membrane vesicles, ribosome, polysaccharide, lipopolysaccharide (LPS), protein-based formulations, etc.) have been developed to date, but none have reached the commercial drug stage [4]. Among the various vaccination strategies currently available for gram-negative bacterial infections, the use of bacterial ghosts (BGs) has shown very promising protective effects. BGs are empty envelopes of bacteria retaining a three-dimensional cellular structure which are produced either genetically or chemically [5]. Developed as safer alternative-killed vaccines, BGs act as efficient carriers for antigens, DNA vaccines, adjuvants, and dendritic cell enhancers. To date, such bacteria as *E. coli*, *Salmonella* spp., *Vibrio* spp., *Pasteurella* spp., *Helicobacter pylori*, *Actinobacillus pleuropneumoniae*, *Aeromonas* spp. [5], *Bordetella bronchiseptica* [6], *Yersinia enterocolitica* [7], *Brucella suis* [8], *Brucella canis* [9], and *Yersinia pestis* [10] have been successfully used to produce engineered BGs. To estimate the protective potency of the most common *K. pneumoniae* serovar O1:K2 [11,12] ghost vaccine, we exploited previously designed lysis plasmids with the φX174 lysis gene *E* and holin–endolysin systems of λ or L-413C phages [13].

## 2. Materials and Methods

### 2.1. Bacterial Strains, Plasmids, and Culture Conditions

The bacterial strains and plasmids used in this study are summarized in Table 1. *K. pneumoniae* strains were incubated at 28 and 37 °C in Luria–Bertani broth (LB) or LB solidified with 1.2% Bacto Agar (Difco). Chloramphenicol (20 µg/mL, Cm) was added as necessary.

### 2.2. Animals

For immunization, we used seven-week-old male and female outbred mice (Lab Animals Breeding Center, Stolbovaya, Moscow Region, Russia). All animal experiments were approved by the Bioethics Committee of the State Research Center for Applied Microbiology and Biotechnology (Permit No: VP-2024/3). This study was performed in strict accordance with the NIH Animal Welfare Insurance #A5476-01 issued on 2 July 2007 and the European Union guidelines and regulations on the handling, care, and protection of laboratory animals (https://environment.ec.europa.eu/topics/chemicals/animals-science_en (accessed on 12 December 2024)).

### 2.3. Construction of K. pneumoniae Strain

Lytic plasmids were transformed into *K. pneumoniae* KPI1627 strain by electroporation.

### 2.4. Lysis Efficiency Assay

*K. pneumoniae* strains were cultured overnight at 28 °C for bacterial ghost (*KP*-BGs) generation. Plasmid pEYR’ lacking the lysis genes was used in control experiments. The induction of lysis was achieved by shifting the temperature from 28 °C to 42 °C when the culture was grown to OD_550_ = 0.6, and the procedure was monitored by the optical densities. Lysed cells were harvested by centrifugation at 5000× *g* for 15 min and washed three times with cold distilled water. The resulting ghosts were named KPI-E, KPI-YK, KPI-EYK, KPI-SRRz, and KPI-ESRRz, and the lysis rate was detected by counting CFUs as described previously [7].

### 2.5. Transmission Electron Microscopy (TEM)

KPI-E, KPI-YK, KPI-EYK, KPI-SRRz, and KPI-ESRRz BGs were subjected to transmission electron microscopy for structural analysis, as we described previously [10].

### 2.6. Production of K. pneumoniae KPI1627/pEYR’-Y-K Bacterial Ghosts

KPI-YK BGs were manufactured in a 10-l fermenter (Yupiter, Solaris, Italy) as previously described [14]. Bacterial cell’s YK-mediated lysis was induced by a temperature change from 28 °C to 42 °C. The lysis rate was detected by OD_550_ and counting of CFUs. The BGs were harvested by centrifugation and washed three times with cold distilled water. The final BG pellet was resuspended in 20 mL of distilled water, freeze-dried, and lyophilized.

### 2.7. Animal Immunization

A total of 60 mice were randomly divided into four groups. Two groups of mice were immunized subcutaneously (s.c.) or intraperitoneally (i.p.) with 0.2 mL of 10^8^ ghost cells of KPI-YK in sterile PBS (pH 7.2). The other two groups of mice served as negative controls and were given s.c. or i.p. phosphate-buffered saline (PBS). Immunization was performed on days 0, 10, and 20. The weight of each animal was monitored daily. The sera of five rodents from each group were taken on 0 h and on day 28 after the initial immunization, and animals were sacrificed for splenic-T cell analysis.

### 2.8. Animal Exposure to Virulent K. pneumoniae KPI1627 Challenge

Ten days after the last immunization, mice (10/group) were administrated by i.p. injection with 0.2 mL of a bacterial suspension (10^4^ CFU) of hypervirulent *K. pneumoniae* KPI1627 strain grown overnight at 37 °C. To confirm the actual number of bacteria, the final suspension was diluted and plated on agar medium. All infected animals were observed over a 14-day period.

### 2.9. Immune Response Assays

#### 2.9.1. ELISA

Serum IgM, IgG, and IgA levels of *K. pneumoniae* BGs were measured by indirect enzyme-linked immunosorbent assay. Briefly, 96-well plates were coated by adding BG (0.1 mg/mL) and incubated overnight at 4 °C. We used goat anti-mouse IgM (1:50,000, Yeasen Biotechnology, Shanghai, China), IgG-HRP (1:5000, Sigma, Saint-Louis, MO, USA), and IgA-HRP (1:25,000, Saint-Louis, MO, Sigma, USA) were used as the detection antibodies. The reactions were developed with TMB (3,3′,5,5′-Tetramethylbenzidine) and stopped with 2 M H_2_SO_4_. The absorbance was measured at 450 nm using a Multiscan FC (Thermo Fisher Scientific, Waltham, MA, USA).

#### 2.9.2. Cellular Responses: Analysis of Stimulated Splenocytes

Cultured mouse splenocytes (10^6^), harvested at day 10 after the third immunization with KPI-YK BGs or PBS, were stimulated with KPI-YK BGs (10^6^ CFU/mL) or concanavalin A (ConA) (5 μg /mL, Sigma-Aldrich, Saint Louis, USA). As a negative control, spleen cells without any activation were used. The culture supernatants were harvested after 48 h. IFN-γ, TNF-α, and IL- 6 levels were measured by ELISA using the Mouse INF-γ ELISA Kit (#558258), Mouse TNF-α ELISA Kit (#560478), and Mouse IL- 6 ELISA Kit (#550950), all according to the manufacturer’s instructions (BD Biosciences, San Jose, CA, USA).

#### 2.9.3. Surface Marker Analysis by Flow Cytometry

Cells were collected after 48 h of recall stimulation with *K. pneumoniae* KPI1627 BGs (10^6^ CFU/mL) and stained with conjugated monoclonal antibodies (mAbs) against cell surface markers for 30 min at 4 °C. The following mAb conjugates were used (BD Biosciences, San Jose, CA, USA): CD3-FITC (clone 17A2), CD4-APC (clone RM4-5), CD8-PE (clone 53-6.7), CD19-APC (clone 1D3), and CD69-BV421 (clone H1.2F3). The samples were washed twice with PBS containing 2% fetal bovine serum. Finally, the samples were resuspended in 150 μL of 4% formaldehyde solution and analyzed using a BD FACSAria III flow cytometer (BD, Franklin Lakes, NJ, USA) with FACSDiva software 8.0.

### 2.10. Statistics

The determination of specific antibodies and cytokines was carried out 3 times for reproducibility, and the results are summarized as means ± standard error of the mean (SEM). Statistical significance was determined by t-tests for unpaired samples and ANOVA. The results from the vaccinated groups were compared with those of the unvaccinated PBS control group; statistically significant comparisons were those with *p* < 0.05. The graphs were prepared using GraphPad Prism version 8.0.0 software for Windows (GraphPad Software, San Diego, CA, USA). The survival curves were performed using the log-rank (Mantel–Cox) test. A *p*-value below 0.05 was considered significant.

## 3. Results

### 3.1. Generation and Characterization of K. pneumoniae Bacterial Ghosts

The lysis plasmids were transformed into the *K. pneumoniae* KPI1627 strain to obtain BGs. The OD_550_ of *K. pneumoniae* KPI1627 cultures was monitored for a period of 2 h after shifting the temperature from 28 °C to 42 °C (Figure 1A). The OD_550_ of *K. pneumoniae* KPI1627/pEYR’ vector control, *K. pneumoniae* KPI1627/pEYR’-E, *K. pneumoniae* KPI1627/pEYR’-S-R-Rz, and *K. pneumoniae* KPI1627/pEYR’-E-S-R-Rz cultures increased constantly after a shift in temperature. Cell viability also increased after lysis induction (Figure 1B). The OD_550_ of *K. pneumoniae* pEYR’-Y-K bacterial culture decreased 60 min after Y-K-lysis induction. For *K. pneumoniae* KPI1627/pEYR’-E-Y-K strain, maximum lysis occurred at 2 h. The lysis rates of KPI-Y-K and KPI-E-Y-K BGs was counted as 99.99% ± 0.01% when the *KP*-BGs were harvested 2 h after the induction. The *K. pneumoniae* KPI1627/pEYR’-Y-K strain, which gives maximum lysis in the shortest possible time after induction, was used to produce KPI-Y-K BG by batch cultivation.

The formation of *K. pneumoniae* ghosts and release of cell contents were confirmed under the transmission electron microscope (Figure 2B–F) by comparing with the *K. pneumoniae* KPI1627/pEYR’ carrying the empty cloning vector (Figure 2A). In total, 93% of untreated *K. pneumoniae* KPI1627/pEYR’ cells had a fine ultrastructure typical for gram-negative bacteria. Untreated cells appeared deep black after staining due to the presence of DNA, protein, and other cell contents. The components of the cell wall were clearly visible. The remaining 7% of bacteria had irreversible structural damage to the cell walls, cytoplasm, and nucleoids.

The lytic action of E protein alone (Figure 2B) resulted in a marked but not complete reduction in electron-optical density of the intracellular content in *K. pneumoniae* KPI1627/pEYR’-E cells compared with wild-type bacteria (Figure 2A). The peripheral areas of the cytoplasm subjected to destruction and lysis are visualized in most bacteria of this sample. The microbial population of this sample contained 35% cells with an intact ultrastructure; the remaining cells had irreversible damage in the form of partial destruction of peripheral areas of the cytoplasm.

The rates of BG formation were highest after lysis was induced by the pEYR’-Y-K (Figure 2C) or pEYR’-E-Y-K (Figure 2D) plasmids when the bacteria completely displaced the cytoplasm to form BG.

The transmission electron microscopy images showed that the *K. pneumoniae* ghosts pEYR’-S-R-Rz (Figure 2E) and pEYR’-E-S-R-Rz (Figure 2F) had irreversible ultrastructural damage such as rupture or disintegration of the cytoplasmic membrane, focal destruction of the cytoplasm and nucleoid, partial or complete excretion of the cytoplasm. The percentage of cells with an intact structure was approximately 42% or 52%, respectively.

### 3.2. Humoral Immune Responses

The titers of IgM, IgG, and IgA antibodies specific to KPI-YK in mice sera were determined after immunization (Figure 3). No specific antibodies were detected in the PBS controls. The IgM and IgG titers were higher after i.p. immunization than in the case of s.c. administration of KPI-Y-K BGs (*p* < 0.05). The highest IgM and IgG titers in mice induced by i.p. were 4000. The IgG titers of the s.c. immunized mice were 500-fold higher than the IgM titers (*p* < 0.05). The levels of IgA in sera of i.p. immunized mice at day 28 post-immunization were higher than IgA of s.c. immunized mice, but no significant differences were observed.

### 3.3. Cytokine Analysis

The concentrations of IFN-γ, IL-6, and TNF-α in the different groups are summarized in Figure 4. The PBS-treated groups showed an inability to induce cytokine production. Analysis of the secretion of cytokines, including IFN-γ, IL-6, and TNF-α, by splenic lymphocytes after stimulation by BGs showed that the levels were all higher than those in the control groups. After immunization with KPI-Y-K BGs, there were statistically significant increases (*p* ≤ 0.05) in IFN-γ, IL-6, and TNF-α levels in the group vaccinated via the i.p. route when compared to the control group. S.c. immunization with KPI-Y-K BGs induced a statistically significant increase (*p* ≤ 0.05) in IL-6 and TNF-α levels when compared to the control group.

### 3.4. The Expression of CD69 on CD3^+^CD4^+^, CD3^+^CD8^+^, and CD19^+^ Cell Subsets

To characterize the functionality of the T/B lymphocytes, we first stimulated mouse splenocytes with *K. pneumoniae* KPI1627 BGs (10⁶ CFU/mL) and proceeded to assess the expression of the CD69 activation marker (Figure 5). In the case of s.c. immunization with KPI-Y-K, CD69 expression was observed on B cells (CD19^+^), but not on T cells (CD3^+^CD4^+^ and CD3^+^CD8^+^). Cytometric analysis showed that the expression levels of CD69 on both T-helper (CD3^+^CD4^+^) and cytotoxic T-lymphocyte (CD3^+^CD8^+^) cells were significantly higher in the i.p.-immunized group compared to the control group (*p* < 0.05).

### 3.5. Protection of Immunized Animals Against K. pneumoniae Challenge

No unusual behaviors were observed in the vaccinated animals, and no significant variance in weight was found between the rodents from different groups.

Ten days after the last immunization, mice from each group were i.p.-infected with 10^4^ CFU of *K. pneumoniae* KPI1627, which is 5000-fold the minimal lethal dosage. All mice injected with PBS died before the sixth day post-challenge. On the contrary, 100% survival of both s.c.- and i.p.-immunized KPI-EY BG-treated groups was observed (Figure 6).

## 4. Discussion

The emergence of multidrug-resistant, hypervirulent *K. pneumoniae* strains requires the development of alternatives to antibiotic therapy that can effectively reduce the incidence of *K. pneumoniae* infections. A search for information on the elaboration of vaccine candidates showed that, despite the use of a wide range of concepts and technologies [15,16,17], no vaccine has been licensed by the WHO to prevent *K. pneumoniae* infection thus far. To date, there are few publications concerning the creation of *K. pneumoniae* bacterial ghosts and their ability to induce potent protective immunity [18]. In our opinion, this is due to the insufficient efficiency of the formation of bacterial ghosts under the influence of bacteriophage φX174 protein E. The main goal of our current study was to determine the efficiency of different bacteriophage lytic enzymes in *KP*-BG generation.

The peptidoglycan is a dynamic bacterial cell constituent involved in many aspects of microbial physiology. It defines bacterial shape, size, division, resistance to osmotic stress, and many other properties responsible for adaptation to dissimilar environmental stresses. All these characteristics are ensured by changes in its structure and, accordingly, sensitivity to the action of various hydrolytic enzymes [19]. Not surprisingly, the plasmids we constructed differed in their lytic activities against different bacterial species. This must be taken into account when creating BGs based on new bacterial species.

*K. pneumoniae* ghosts, a previously constructed set of plasmids, were used, which have been successfully used for the design and production of *E. coli* [13] and *Y. pestis* BGs [10]. It should be noted that inclusion of the holin (K) genes into the lytic plasmid, instead of protein E, significantly increased the efficiency of BG formation. More precisely, they were turned not into BGs that retained the three-dimensional form of the original microorganisms, but into structures more similar to shapeless sacs. The loss of rigidity by cells can be explained by the total hydrolysis of peptidoglycan, which served as the cell carcass. The growth curves of KPI-E, KPI-YK, KPI-EYK, KPI-SRRz, and KPI-ESRRz also confirmed a significantly higher lysis efficiency in strains expressing the holin and endolysin genes.

As can be seen from transmission electron micrographs, the expression of only protein E in *K. pneumoniae* cells led to a noticeable, but not complete, lightening of the intracellular contents compared to wild-type bacteria. It seems that the outflow of contents from the cells was hampered either by a decrease in the permeability of the intermembrane tunnels or by an increase in the viscosity of the cytoplasm. The introduction into *K. pneumoniae* cells, instead of the E protein, of alternative factors that lyse the bacterial cell wall by hydrolyzing the peptidoglycan leads to the loss of the murein skeleton. The resulting frameless cell envelopes are more reminiscent of bacterial sacs or bacterial skins than BGs. Although such structures are less naive than classical bacterial ghosts, they induce effective protection against infection by a hypervirulent strain of *K. pneumoniae* and can be recommended as candidate vaccines. The variant of bacterial sac formation chosen by us positively correlated with the increase in protective activity in *Y. pestis* [10] and *Salmonella* [20]. This was probably due to the complete hydrolysis of murein, which has been shown to have immunosuppressive properties. In our experiments, we did not compare the protective potency of bacterial ghosts and bacterial sacs due to the fact that we were unable to significantly increase the content of bacterial ghosts to more than 65%, and 35% of cells remained without visible changes. However, in our opinion, it would be appropriate to compare the immunogenic activity of ghosts and sacs on other types of bacteria.

The co-expression of holin and endolysin lysis genes allowed for rapid and highly efficient inactivation of the hypervirulent *K. pneumoniae* strain during the ghost formation procedure. TEM showed that the bacteria had lost their original shape, becoming empty sacs devoid of cytoplasmic components. Based on these data, it was decided to evaluate the immunogenic activity and protective properties of the strain harboring phage L-413C genes encoding holin and endolysin.

Administration routes may considerably change vaccine efficiency due to the local microenvironment variance at the inoculation site [21]. This study assessed the effectiveness of vaccination of the tested bacterial ghosts prepared on the basis of a hypervirulent clinical isolate of *K. pneumoniae* in the formation of a humoral and cellular immune response upon s.c. and i.p. administration to outbred mice. Specific IgG and IgA antibodies were detected in sera of the s.c. group treated with KPI-Y-K BGs at day 28 after the first immunization. The i.p. group exhibited high levels of specific IgM, IgG, and IgA antibodies. The s.c. group had significantly higher IgG antibody levels and significantly increased IgA levels, while the i.p. group had elevated amounts of IgM and IgG antibodies and also produced high levels of IgA, showing that both the s.c. and i.p. routes induced potent humoral immune responses.

The interaction of multiple pro-inflammatory cytokines forms the primary network of host resistance to *K. pneumoniae*, including interferon (IFN-γ), interleukin-6 (IL-6), and tumor necrosis factor (TNF-α) [21,22]. The deficiency and neutralization of TNF-α consistently resulted in increased mortality due to impairment of lung antibacterial host defense and/or the dysregulation of cytokine production, as was observed in the mouse model [22,23,24]. Interferon gamma (IFN-γ) is a key regulator of the host defense system and is primarily associated with inflammation- and cell-mediated immune responses. IFN-γ induces the rapid production of reactive nitrogen species (RNS) and reactive oxygen species (ROS), as well as the generation of antimicrobial peptides that contribute to bacterial killing. In addition, IFN-γ induces the production of cytokines that are responsible for the activation of other immunocompetent cells. Intratracheal inoculation of IFN-γ knockout mice with *K. pneumoniae* significantly increased the mortality rate. This was accompanied by a 100-fold increase in bacteria in the lung within two days of infection [25]. The IL-6^−/−^ mice intranasally or intraperitoneally inoculated with *K. pneumoniae* were less likely to survive than wild-type controls, and at the time of death, they had higher numbers of bacteria [26]. Our results demonstrated that splenic lymphocytes from i.p.-immunized mice after re-stimulation with KPI-Y-K BGs elicited significantly higher levels of IFN-γ, TNF-α, and IL-6 compared with the control group. On the other hand, there was no significant difference in the level of IFN-γ secretion by splenic lymphocytes from s.c.-immunized mice compared with the control group.

Bacterial ghosts can target antigenic components to antigen-presenting cells and can also be internalized by APC or epithelial cells [27]. Kudela et al. [28] reported that epithelial cells strongly internalized BGs independently of the bacterial species. This means that BGs could activate both MHC-I and MHC-II antigen-processing and presentation pathways, promoting the activation of CD4+ and CD8+ T cells [29]. According to our results, 24-h stimulation of lymphocytes obtained from mice i.p. immunized with KPI-Y-K BGs resulted in the activation of (CD69+) CD4+ and CD8+ T cells. This indicates the development of a targeted T-cell immune response to *K. pneumoniae* antigens. Additionally, a population of B lymphocytes was observed, which enhanced the expression of CD69+ under the influence of *K. pneumoniae* antigens in vitro. The s.c. immunization of mice resulted in the generation of B lymphocytes, which were activated by *K. pneumoniae* antigens. In contrast, T lymphocytes exhibited no reactivity toward the same antigens.

However, differences in the humoral and cellular immune responses between s.c.- and i.p.-injected groups were not principal for death prevention, because two routes of administration of KPI-Y-K BGs led to complete protection of all animals against i.p. *K. pneumoniae* infection.

## 5. Conclusions

The presence of protein E is not sufficient to guarantee the generation of genetically engineered *K. pneumoniae* ghosts. At the same time, a combination of holin and endolysin from phage L-413C formed frameless peptidoglycan-free ghosts, ensuring reliable protection against death for mice infected with a hypervirulent *K. pneumoniae* strain. Peptidoglycan is not required for the induction of intense immunity by the *K. pneumoniae* bacterial ghost vaccine. The frameless cell envelopes are more reminiscent of bacterial sacs or bacterial skins than BGs. Although the structural components of our proposed vaccine based on the bacterial cell envelops are less naive than classical BGs, they provide effective protection against infection with a hypervirulent strain of *K. pneumoniae,* and this can be recommended as a candidate vaccine.

These results are another promising step toward demonstrating the clinical potential of *K. pneumoniae* ghost vaccines to induce a potent immune response.

## Figures and Tables

**Figure 1 vaccines-13-00059-f001:**
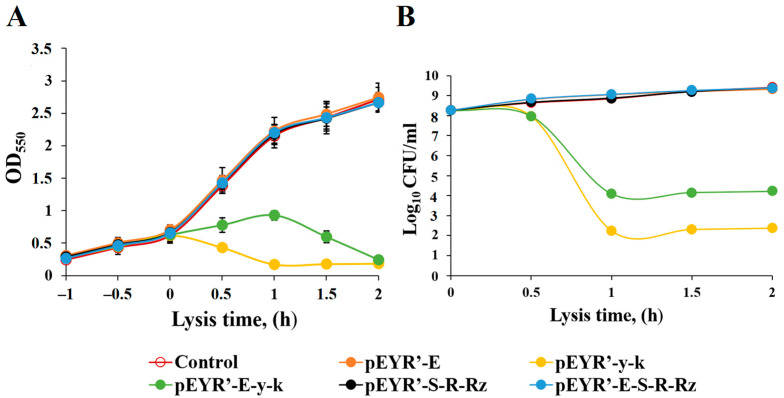
Preparation of *K. pneumoniae* KPI1627 BGs. Growth and lysis were monitored by measuring OD_550_ (**A**) and the determination of the number of CFU (**B**). The data are presented as the mean ± s.d. of three samples.

**Figure 2 vaccines-13-00059-f002:**
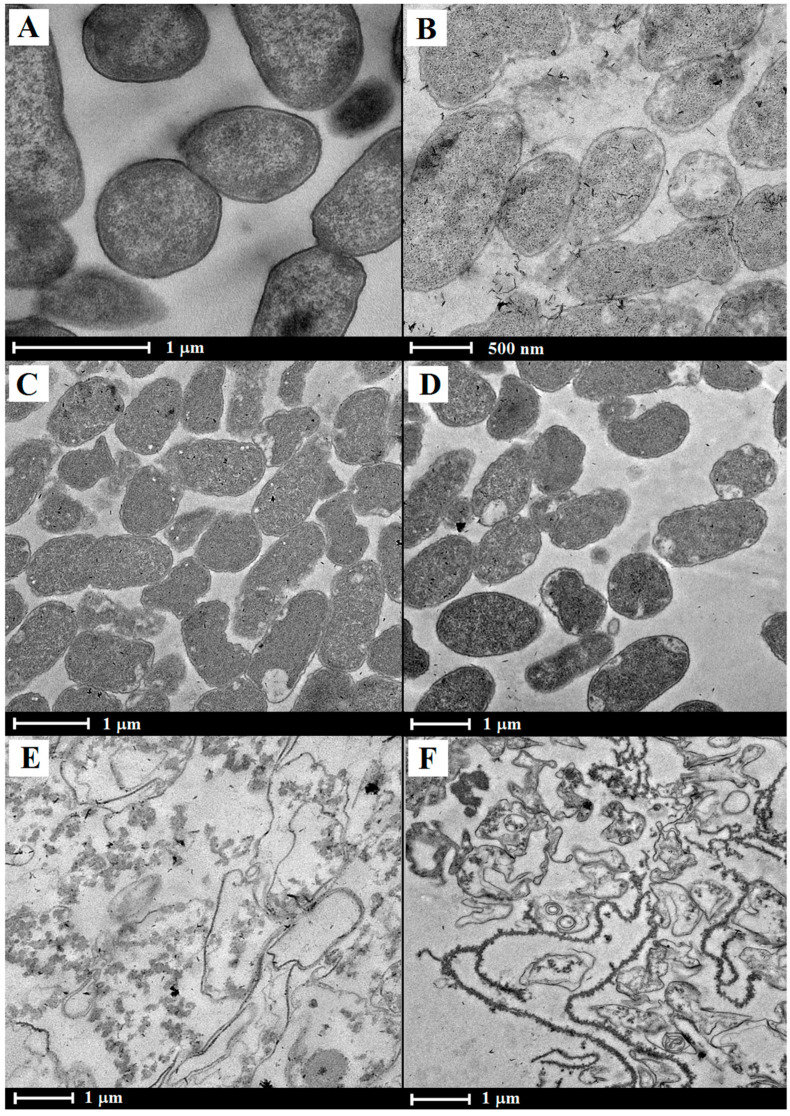
Transmission electron micrographs of *K. pneumoniae* strains: (**A**) KPI1627, (**B**) KPI1627/pEYR’-E, (**C**) KPI1627/pEYR’-S-R-Rz, (**D**) KPI1627/pEYR’-E-S-R-Rz, (**E**) KPI1627/pEYR’-Y-K, (**F**) KPI1627/pEYR’-E-Y-K. The bar represents 1 μm (**A**,**C**–**F**) or 500 nm (**B**).

**Figure 3 vaccines-13-00059-f003:**
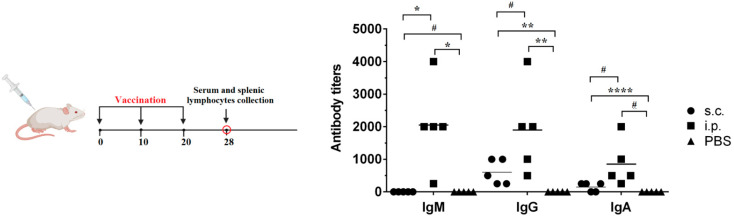
Antibody response in sera of mice immunized s.c. and i.p. with KPI-Y-K and PBS. #—*p* > 0.05; *—*p* < 0.05; **—*p* < 0.005; ****—*p* < 0.0001.

**Figure 4 vaccines-13-00059-f004:**
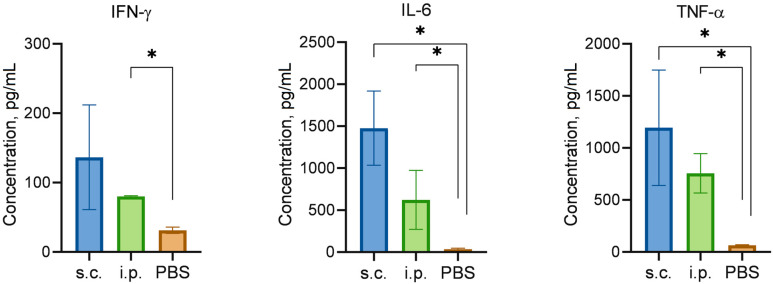
Specific IFN-γ, IL-6, and TNF-α levels of splenic lymphocytes from immunized mice. * *p* < 0.05 vs. PBS group.

**Figure 5 vaccines-13-00059-f005:**
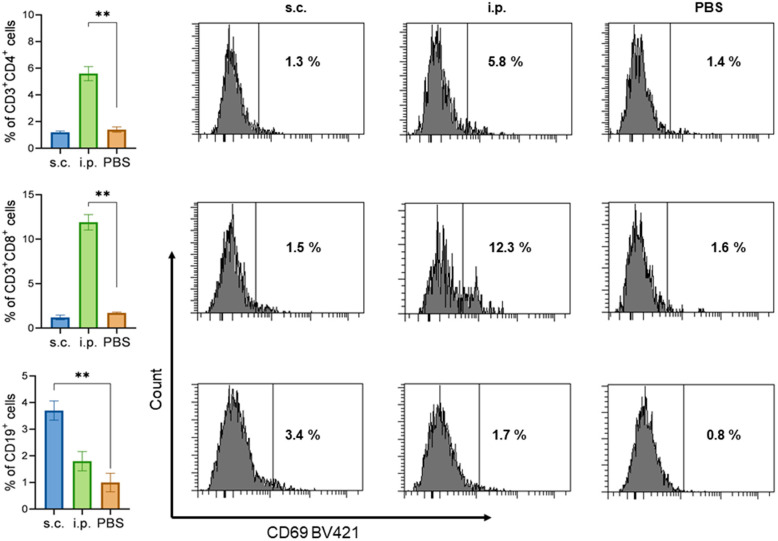
The expression levels of CD69 on CD3^+^CD4^+^, CD3^+^CD8^+^, and CD19^+^ cell subsets of splenic lymphocytes from immunized mice. The splenic lymphocytes of mice were separated 28 days after the first immunization, and corresponding BGs were used as immunogens. Following a 48-h incubation period, lymphocytes were harvested and subjected to flow cytometry analysis. ** *p* < 0.005 vs. PBS group. Graphs and histograms show the distribution of CD69 expression in the lymphocyte subsets.

**Figure 6 vaccines-13-00059-f006:**
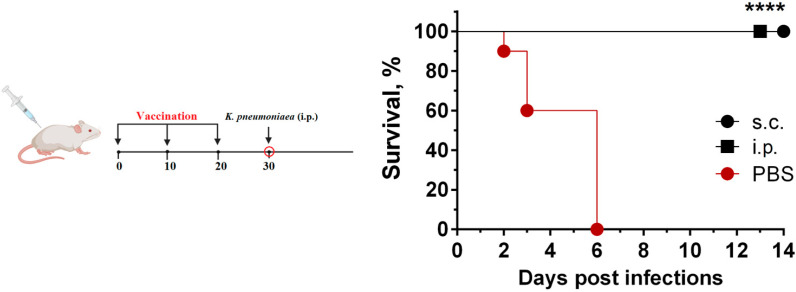
Protection of *KP*-BGs against a lethal challenge with the wild-type *K. pneumoniae* KPI1627 strain. Mice were subjected to i.p. and s.c. immunization with KPI-YK BGs at day 0 and boosted twice at 10 and 20 days. Ten days after the last immunization, 10 mice from each group were challenged i.p. with 10^4^ CFUs of *K. pneumoniae* KPI1627 (5000 LD_50_). **** *p* < 0.0001.

**Table 1 vaccines-13-00059-t001:** Bacterial strains and plasmids used in this study.

Strains/Plasmid/Bacteriophage	Relevant Genotype or Annotation	Source
*K. pneumoniae*		
KPI1627	Hypervirulent, hypermucoviscous, serotype O1:K2 (LD_50_ for mice ≤ 2 CFU)	SCPM-O ^1^
KPI1627/pEYR’	KPI1627 containing pEYR’	This study
KPI1627/pEYR’-E	KPI1627 containing pEYR’-E	This study
KPI1627/pEYR’-E-Y-K	KPI1627 containing pEYR’-E-Y-K	This study
KPI1627/pEYR’-Y-K	KPI1627 containing pEYR’-Y-K	This study
KPI1627/pEYR’-E-S-R-Rz	KPI1627 containing pEYR’-E-S-R-Rz	This study
KPI1627/pEYR’-S-R-Rz	KPI1627 containing pEYR’-S-R-Rz	This study
**Plasmids**		
pEYR’	Expression vector, phage Lambda modified right promoter (pR’) (Cm^r^)	[13]
pEYR’-E	Lysis plasmid, pEYR’-lysis E (Cm^r^)	[13]
pEYR’-E-Y-K	Lysis plasmid, pEYR’-lysis E, Y, K (Cm^r^)	[13]
pEYR’-Y-K	Lysis plasmid, pEYR’-lysis Y, K (Cm^r^)	[13]
pEYR’-E-S-R-Rz	Lysis plasmid, pEYR’-lysis E, S, R, Rz (Cm^r^)	[13]
pEYR’-S-R-Rz	Lysis plasmid, pEYR’-lysis S, R, Rz (Cm^r^)	[13]

^1^ The State Collection of Pathogenic Microbes and Cell Cultures on the base of the State Research Center for Applied Microbiology and Biotechnology (“SCPM-Obolensk”).

## Data Availability

All the data will be provided on reasonable request.

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
