# Peer review of "Genetically Engineered Bacterial Ghosts as Vaccine Candidates Against Klebsiella pneumoniae Infection"

_vaccines, 2025, doi:10.3390/vaccines13010059_

Round 1
Reviewer 1 Report
Comments and Suggestions for Authors
Dear authors
I have read your manuscript Genetically engineered bacterial ghosts as vaccine candidates against Klebsiella pneumoniae infection, and these are my comments and suggestions:
General comment
Lines 250-252: Here you describe that all mice injected with the genetically engineered bacterial ghosts survived the bacterial challenge whilst none of the control mice did. I would encourage you to put this more prominently to additional sections of your manuscript (such as the abstract, discussion, or conclusions), mentioning the number of mice used etc. You may also consider changing the title to Genetically engineered bacterial ghosts show effective protection against Klebsiella pneumoniae in the mouse model. It is usually good not to be too bold in science. Nevertheless, I think you could point out your success a bit more in this manuscript.
Specific comments
Line 30: Omit "In animal model".
Lines 43-45: Change to "Multidrug resistance in K. pneumoniae renders antibiotic therapy ineffective, highlighting the need for the development of Klebsiella vaccines".
Line 57: What are "alternative-killed vaccines"?
Line 62: Omit ", which is producedlly".
Line 70: Why was chloramphenicol added in some cases?
Line 251: Change "immunized with PBS" to "injected with PBS" or similar.
Line 260: Change "alternative" to "alternatives" and omit "measures".
Lines 264-266: Omit the sentence "Most studies devoted to development of K. pneumoniae candidate vaccines with the help of different technological platforms describe in detail the materials and methods used in these investigations as well as their results."
Lines 266-271: This sentence is confusing and too long in my opinion. Please reformulate.
Author Response
Dear reviewer, thank you very much for your thorough analysis and high evaluation of our manuscript. We have substantially revised it and made the following improvements and corrections in accordance to the requests:
Comment 1: Line 30: Omit "In animal model".
Response 1: Done.
Comment 2: Lines 43-45: Change to "Multidrug resistance in K. pneumoniae renders antibiotic therapy ineffective, highlighting the need for the development of Klebsiella vaccines".
Response 2: Done.
Comment 3: Line 57: What are "alternative-killed vaccines"?
Response 3: "Alternatively killed vaccines" are vaccines obtained by inactivation of bacteria using lytic enzymes of bacteriophages, rather than classical chemical inactivation methods, e.g., formaldehyde.
The development of ghost vaccines trials
AM Batah, TA Ahmad - Expert review of vaccines, 2020 - Taylor & Francis
… vaccines, as well as, being efficient alternative-killed vaccines. The coming years will show
more … Ghost-vaccine techniques enable the production of safer alternative-killed vaccines, …
Role of immunobiotic lactic acid bacteria as vaccine adjuvants
M Dadar, Y Shahali, N Mojgani - … in the Prevention and Management of …, 2022 - Elsevier
… Ghost-vaccine techniques enable the production of safer alternative-killed vaccines; there
are platforms that express a wide number of antigens and DNA encoding epitopes, carrier …
Comment 4: Line 62: Omit ", which is producedlly".
Response 4: Done.
Comment 5: Line 70: Why was chloramphenicol added in some cases?
Response 5: For lytic plasmid retention.
Comment 6: Line 251: Change "immunized with PBS" to "injected with PBS" or similar.
Response 6: Done.
Comment 7: Line 260: Change "alternative" to "alternatives" and omit "measures".
Response 7: Done.
Comment 8: Lines 264-266: Omit the sentence "Most studies devoted to development of K. pneumoniae candidate vaccines with the help of different technological platforms describe in detail the materials and methods used in these investigations as well as their results."
Response 8: Done.
Comment 9: Lines 266-271: This sentence is confusing and too long in my opinion. Please reformulate.
Response 9: Done.
To date, there are few publications concerning the creation of K. pneumoniae bacterial ghosts and their ability to induce potent protective immunity (18).

Reviewer 2 Report
Comments and Suggestions for Authors
This manuscript [“Genetically engineered bacterial ghosts as vaccine candidates against Klebsiella pneumoniae infection”] by Dentovskaya et al. demonstrates the potential of K. pneumoniae ghost vaccines to induce a potent immune response. This is follow-up work by the authors after their previously published works on Bacterial Ghosts; (1) “Peptidoglycan-Free Bacterial Ghosts Confer Enhanced Protection against Yersinia pestis Infection”, Vaccines (Basel). 2021 Dec 30;10(1):51. doi: https://doi.org/10.3390/vaccines10010051, and (2) “The Efficiency of Bacteriophage Lytic Enzymes in the Course of Bacterial Ghost Generation”, Mol. Genet. Microbiol. Virol. 37, 131–137 (2022). https://doi.org/10.3103/S0891416822030077. To improve the efficiency of Bacterial Ghosts (BGs) formation and to maximize their efficacy (i.e. killing of the bacteria), the authors exploited previously designed plasmids with the lysis gene E from bacteriophage Ï•X174 or with holin-endolysin systems of λ or L-413C phages. The results indicate that co-expression of the holin and endolysin genes leads to more rapid and efficient K. pneumoniae lysis than that mediated by only single gene E or the low functioning holin-endolysin system of λ phage.
In short, the present study is quite extensive and the results are quite promising. I, therefore, recommend it for publication in the journal of Vaccines.
Minor Issues:
1) Figure 1: The figure caption says “The data are presented as the mean ± s.d. of three samples.” However, the error bars are not visible. Please include them. How were these data fitted?
2) The writing is a little sloppy and full of grammatical and typographical errors – please read carefully and correct those.
3) Please try reducing self-plagiarism, e.g. from the previously published manuscripts such as “Peptidoglycan-Free Bacterial Ghosts Confer Enhanced Protection against Yersinia pestis Infection”; Vaccines 2022, 10(1), 51; https://doi.org/10.3390/vaccines10010051.
Author Response
Dear reviewer, thank you very much for your thorough analysis of our manuscript.
Minor Issues:
Comment 1: Figure 1: The figure caption says “The data are presented as the mean ± s.d. of three samples.” However, the error bars are not visible. Please include them. How were these data fitted?
Response 1: As per your recommendation, we have included the ± s.d. option in Figure 1. In the case of the image in Figure 1A (arithmetic scale), we did indeed see standard deviations. However, when using the geometric scale (Figure 1B), they were too small to be visualized. The absence of visible deviations in Figure 1A is explained by their insignificant values.
Comment 2: The writing is a little sloppy and full of grammatical and typographical errors – please read carefully and correct those.
Response 2:
Comment 3: Please try reducing self-plagiarism, e.g. from the previously published manuscripts such as “Peptidoglycan-Free Bacterial Ghosts Confer Enhanced Protection against Yersinia pestis Infection”; Vaccines 2022, 10(1), 51; https://doi.org/10.3390/vaccines10010051.
Response 3: We cannot refuse to cite our previous article because it is the first to discuss the importance of complete hydrolysis of peptidoglycan for the protective activity of bacterial ghosts.
Reviewer 3 Report
Comments and Suggestions for Authors
The authors present a novel vaccine BG candidate to counter K. peumoniae infections, which is based on a method recently developed by the authors to enhance BG formation using genetically engineered K. pneumoniae lysis strains co-expressing the holin and endolysin genes from the L-413C phage.
The authors provide convincing evidence that the obtained frameless cell envelopes elicit an effective immune response in-vivo, offering a sufficient protection to avoid severe disease symptoms.
The study is well conducted leading to sound conclusions. Employing BGs generated by the approach presented may offer an efficient and rather safe vaccination strategy avoiding K. pneumoniae infections.
As an amendment, the prospects to generalise this approach as a safe vaccination strategy against various bacterial infections may be concisely outlined, the more so as many sources of adverse effects such as adding toxic adjuvants etc., may be avoided. In this respect, it may be briefly discussed whether the "bacterial sacs" likely represent a safer immunostimulans than conventional BGs (i.e. causing less adverse effects).
As a minor point, the first paragraph in section 2.9 can be omitted as it is identical to the previous section 2.8.
Author Response
Dear reviewer, thank you very much for your thorough analysis and high evaluation of our manuscript. We have substantially revised it and made the following improvements and corrections in accordance to the requests:
Comment 1: As an amendment, the prospects to generalise this approach as a safe vaccination strategy against various bacterial infections may be concisely outlined, the more so as many sources of adverse effects such as adding toxic adjuvants etc., may be avoided. In this respect, it may be briefly discussed whether the "bacterial sacs" likely represent a safer immunostimulans than conventional BGs (i.e. causing less adverse effects).
Response 1: Text is added:
The variant of bacterial sac formation chosen by us positively correlated with the increase of protective activity in Y. pestis [10] and Salmonella [https://doi.org/10.1038/srep45139] [20]. This was probably due to the complete hydrolysis of murein, which has been shown to have immunosuppressive properties. In our experiments, we did not compare the protective potency of bacterial ghosts and bacterial sacs, due to the fact that we were unable to significantly increase the content of bacterial ghosts to more than 65%, 35% of cells remained without visible changes. However, in our opinion, it would be appropriate to compare the immunogenic activity of ghosts and sacs on other types of bacteria.
Comment 2: As a minor point, the first paragraph in section 2.9 can be omitted as it is identical to the previous section 2.8.
Response 2: Deleted.
Reviewer 4 Report
Comments and Suggestions for Authors
Dear authors,
MDPI endorses the ARRIVE guidelines (arriveguidelines.org/) for reporting experiments using live animals. Authors and reviewers must use the ARRIVE guidelines as a checklist, which can be found at https://arriveguidelines.org/sites/arrive/files/documents/Author%20Checklist%20-%20Full.pdf. The journal Vaccines requires authors to submit the completed checklist at submission, and it will be made available to reviewers.
Editors reserve the right to reject submissions that do not adhere to these guidelines based on ethical or animal welfare concerns, or if the procedure described does not appear to be justified by the value of the work presented“
Would you submit this checklist?
Line 82 provided link is not functional, correct it
Line 121-124 are identical to Line 115-119
References 1 and 2 are 40-years-old. Are still relevant?
LIne 284 check word Inclusion stated twice
References are only 29 and 12/29 are from the latest 5 years. Is your topic of relevance or it is more historical issue?
After upgrading your mauscript I am ready to review it again.
Author Response
Dear reviewer, thank you very much for your thorough analysis of our manuscript. We have substantially revised it and made the following improvements and corrections in accordance to the requests:
Comment 1: Would you submit this checklist?
Response 1: Dear reviewer, the translation of the Russian version of the checklist is enclosed please find.
Comment 2: Line 82 provided link is not functional, correct it
Response 2: Done.
Comment 3: Line 121-124 are identical to Line 115-119
Response 3: Done.
Comment 4: References 1 and 2 are 40-years-old. Are still relevant?
Response 4: The link to the 1985 publication was replaced with a more recent one from 2020. Link #2 was not changed, since it is from 2017.
Comment 5: LIne 284 check word Inclusion stated twice
Response 5: Done.
Comment 6: References are only 29 and 12/29 are from the latest 5 years. Is your topic of relevance or it is more historical issue?
Response 6: We have tried to cite the primary source publication whenever possible to give credit to the discoverers rather than to the researchers who are, to some extent, involved in compiling the data.
